# Social Interaction, Survival Stress and Smoking Behavior of Migrant Workers in China—An Empirical Analysis Using CHARLS Data from 2013–2018

**DOI:** 10.3390/bs13080680

**Published:** 2023-08-12

**Authors:** Fanzhen Kong, Huiguang Chen, Yu Cheng

**Affiliations:** College of Public Administration, Nanjing Agricultural University, Nanjing 210095, China; kongfanzhen2022@njau.edu.cn (F.K.); chengyu6936@163.com (Y.C.)

**Keywords:** smoking behavior, migrant workers, social interaction, work stress, gender difference, regional heterogeneity

## Abstract

Smoking is a major public health problem in most countries and usually occurs in marginalized groups. Analyzing the smoking behavior of migrant workers, a marginalized group in China, is of practical significance. Using panel data from the China Health and Retirement Longitudinal Study (CHARLS) database from 2013 to 2018, this study examined influence factors of smoking behavior (whether to smoke and smoking frequency) among migrant workers in China through the Heckman two-stage model. The results showed that the smoking rates of migrant workers were positively associated with social activity and a sense of loneliness, while smoking frequency was negatively associated with work stress and life satisfaction. Meanwhile, smoking behavior was associated with the demographic variables such as gender, age, and education level. Gender differences in smoking behavior were particularly notable among Chinese migrant workers. Furthermore, there was regional heterogeneity in smoking behavior among migrant workers. Smoking behavior in the eastern region was mainly influenced by psychological factors of wellbeing, such as social activity and life satisfaction, while it was affected by material conditions such as income in the central-western region. Effective strategies to control tobacco use among migrant workers are proposed in order to promote social integration between urban and rural residents, increase vocational education and training, and strengthen anti-smoking propaganda among migrant workers.

## 1. Introduction

As the World Health Organization (2021) pointed out, there were over one billion people smoking globally, which accounted for approximately one-seventh of the world’s population. Tobacco-related diseases result in the death of approximately eight million people each year, leading to a loss of 1.4 trillion US dollars to the global economy [1]. This has a disproportionate impact on people in low-income and middle-income countries. As a result, the World Health Organization has identified tobacco smoking as one of the leading risk factors for premature mortality worldwide [2]. China, as the world’s largest tobacco consumer and producer, leads the world in both the consumption of tobacco and smoking-related deaths [3,4]. The Chinese government has actively promoted tobacco control significantly via measures such as joining the Framework Convention on Tobacco Control (FCTC) proposed by the World Health Organization in 2003, promulgating stricter regulations on smoking control in public places in 2014 to stipulate that indoor public places be strictly non-smoking areas, and formulating the Healthy China Action (2019–2030) to list tobacco use as one of the major special actions in 2019 [5]. However, according to the China Adult Tobacco Survey Report 2018 from the Chinese Center for Disease Control and Prevention [6], the smoking rate among those aged 15 and above in China was 26.6%, with 50.5% of males and 2.1% of females smoking. This was substantially higher than the global smoking rate of 19.2% during the same period. Moreover, marginalized populations are disproportionately affected by smoking globally [7]. This is particularly true of migrant workers, who are more likely to generate uncertainty in management. Thus, the smoking control situation in China is highly complex and severe, and the pathway to tobacco control remains a considerable challenge.

China is a large agricultural country and the rural resident population was 509.79 million in 2021, accounting for 36.11% of the total population [8]. Over the past several years, with the rapid modernization, industrialization and new urbanization of China, a large-scale migration of the rural population to urban areas has occurred. Among these people, as a special floating population in China, more and more migrant workers have chosen to go out across counties or even provinces to seek new job opportunities. The term “rural-to-urban migrant” in China refers to farmers who move from rural to urban areas in search of employment and a better living standard without obtaining permanent urban residency or hukou [9]. Due to China’s household registration (hukou) system, it is difficult for migrant workers to obtain the same social welfare opportunities as urban residents, which is reflected in housing subsidies, pension plans, their children’s education, and so on. Moreover, migrant workers are in a state of limbo, hovering on the edge of urban and rural areas. In terms of job selection, migrant workers mainly work in the low-skilled labor market. In terms of social interaction, they mainly interface with fellow townsmen and colleagues every day, which makes it difficult for them to integrate into the lives of urban residents. Meanwhile, these multiple pressures, along with unstable living and employment conditions, high mobility, and social discrimination, easily lead to risky behavior with adverse health outcomes, such as smoking [10].

Previous studies have shown that cigarette smoking and alcohol drinking are prevalent behaviors among migrant workers and that over 60% of the groups have a history of smoking or drinking, with the majority being in their prime years [11]. It has also been shown that there are obvious gender differences [7,10]. The smoking behavior of rural–urban migrants is associated with unstable living and working conditions [12,13]. Compared with other groups, the migrant worker group is prone to be in a state of psychological anxiety because of their high work intensity, low income and poor job stability, factors that can probably lead to depression [14]. In addition, vulnerability to social discrimination and a lack of social integration and acceptance from urban residents leads to the loss of migrant workers’ sense of location and belonging, ultimately worsening the unhealthy risks facing this cohort. As several studies have examined, work-related pressures have increased, such as worse working conditions, higher working intensity, and longer working time. Additionally, it has been shown that migrants experience more social discrimination and inequality in daily life. Therefore, the probability of their engaging in the unhealthy behavior of smoking has increased significantly [15,16,17]. An empirical analysis based on the data of migrant workers in Shanghai showed that the smoking risk experienced by migrant workers in the building trades was greatly increased due to high-pressure work and payment often being delayed or withheld [18,19]. Through comparing returning migrants and non-migrants in Mexico, it was found that the health status of returning migrants was worse in terms of obesity, smoking, and mental health [20]. Socioeconomic status (SES) is usually a multidimensional construct involving education, occupation and income [21], and it is widely regarded as being an important determinant of smoking behavior. Previous studies have shown that SES had different effects on smoking behavior in countries with different income levels, groups of different ages, and different genders, with significant regional differences and gender effects [22]. Weyers et al. determined that low SES was closely related to unhealthy behaviors using cross-sectional data from population-based epidemiological studies in Germany [23]. However, on the basis of cross-sectional data from Beijing, Chen et al. found that smoking level was positively associated with age, education, income, and number of jobs held, negatively associated with work intensity, life satisfaction and living with relatives in Beijing [24]. Overall, survival stress and maladaptation to urban life are probably key determinants of smoking behavior [25,26,27]. These studies provide rich research foundations and practical experience to assist in further analysis.

Despite the numerous studies conducted by scholars, both domestically and internationally, on the smoking behavior of migrant workers or transnational migrants and its influencing factors, the existing literatures are often based solely on the use of certain cross-sectional data to measure the smoking behavior of migrant workers via the use of either a comprehensive index or a single dimension. There is still further space to explore by combining multiple dimensions. Therefore, this study used panel data from the China Health and Retirement Longitudinal Study (CHARLS) database over the period of 2013–2018 to analyze the smoking status, gender difference, influence degree and regional differences of migrant workers in China from the perspectives of social interaction, work stress and life stress. The main concepts and framework structures are displayed in Figure 1. There are three potential contributions of this paper. Firstly, we examined the impact of social interaction, work stress and life stress on their smoking behavior, including whether to smoke and smoking frequency, combined with the reality faced by migrant workers. Secondly, we used multi-period unbalanced panel data from the CHARLS database for 2013–2018 to analyze smoking behavior by means of expanding the sample size while increasing comparisons with the results of cross-sectional studies. Thirdly, we paid more attention to the influence of social interaction on the smoking behavior of migrant workers. At present, China is experiencing a transition from an urban–rural dual structure to the development of urban–rural integration, and the protection of the rights and interests of migrant workers as a marginalized and vulnerable group is of particular importance. These research findings maybe provide empirical support for public health policy design and facilitate the effective implementation of tobacco control strategies for migrant workers, a relatively marginalized subgroup of Chinese floating population, from the perspective of social integration, public propaganda and vocational training.

The purpose of this paper was to examine the prevalence, gender difference and influencing factors of smoking behavior, including whether to smoke and smoking frequency. Particularly, we highlighted the effect of social interaction, work stress and life stress on the smoking behavior of migrant workers. The remaining structure of this paper was as follows: the second part introduced the data source, variable selection and model method, and generated descriptive statistics on the data and basic variables; the third part was the result analysis, including regression results, regional heterogeneity analysis and robustness test of the model; the fourth part further deepened and expanded on the research results and their significance; and the fifth part summarized the main conclusions and proposed relevant suggestions for tobacco control measures that target migrant workers.

## 2. Materials and Methods

### 2.1. Data Sources

The data used in this study all come from the China Health and Retirement Longitudinal Study (CHARLS) database. The CHARLS is a set of high-quality micro-survey data comprising Chinese adults aged 45 and over and their family members for scientific research [28]. The baseline national wave of CHARLS was conducted in 2011 by the Social Science Research Center of Peking University, and a stratified (by per capita GDP of urban districts and rural counties) multi-stage (county/district and village/community household) PPS random sampling strategy was adopted. The survey covered 17,000 respondents from about 10,000 households among 150 counties/districts in 28 provinces [29]. Subsequently, these samples were followed up every two or three years, and survey data from four periods in 2011, 2013, 2015 and 2018 has been publicly released. All participants provided informed consent before participating in the investigation. The ethical approval of data collection was provided by the Biomedical Ethics Review Committee of Peking University [30]. All detailed information about the CHARLS can be found at http://charls.pku.edu.cn/en (accessed on 26 June 2023).

In this study, we used the CHARLS database from the 2013, 2015 and 2018 waves to create a non-equilibrium short panel dataset that included migrant workers who completed the follow-up. One of the many attractive features of this dataset, which makes it particularly suitable for our empirical analysis, is that it provides not only rich sample information such as demographic background, employment status, income, health status and other contents, but also provides specific variable information closely related to smoking behavior and social interaction. Due to the age limit of the survey samples in the CHARLS database, we focused in this paper on the subgroup of migrant workers aged 45 or above, with an associate degree or below and a rural household registration, who had worked in non-agricultural jobs for more than three months. According to the samples of migrant workers in 2013, those of 2015 and 2018 matched. After removing unmatched data and missing samples, we finally obtained 1164 data observations covering 388 survey samples suitable for use in tracking investigations over three consecutive waves.

### 2.2. Variable Measurement

#### 2.2.1. Smoking Behavior

Smoking behavior is a progressively developing process and one of the addictive behaviors. According to the FCTC proposed by the WHO in 2003, smoking is defined as a behavior that includes possessing or controlling lit tobacco products, regardless of whether smoke is actually inhaled or exhaled. In this study, we measured the smoking behavior of migrant workers using two variables: one was smoking status, namely whether the interviewee smokes (yes = 1, no = 0); the other was smoking frequency, namely the number of cigarettes smoked per day among smokers. Therefore, the prevalence of smoking among migrant workers can be evaluated (Table 1).

#### 2.2.2. Stress and Social Interaction

Given that stress is a multi-dimensional and general concept, this study designed two stress dimensions, including work stress and life stress, by taking into account the main stress sources faced by migrant workers. Work stress referred to the stress caused by heavy workloads, changes in job positions and high job responsibilities. The main measurement indicators in this study included work stress, working time and income, which corresponded to direct questions in the CHARLS questionnaire such as “What is the after-tax salary including bonus in the last month?” and “How many hours did you work per day on average in the past year, excluding meal breaks but including any paid or unpaid overtime?” To account for life stress, this study considered daily work income and basic expenditures with reference to the minimum living standard, i.e., the international poverty line released by the World Bank in 2022 [31], measured by the relative value of daily income and minimum living expenses. The measurement indicators included the life stress and life satisfaction, referring to the direct question in the CHARLS questionnaire: “How satisfied are you with your life-as-a-whole?” and some indirect indicators. More details on these variables could be found in Table 1 and Appendix Table A1.

The key explanatory variable frequently used was social interaction, including social activity, social number and sense of loneliness. Specific information about social interaction was mainly obtained through the questions in the CHARLS questionnaire, such as “Have you done any of these activities in the last month? (Multiple choices allowed)”, for which there were 12 options, and “I felt lonely?” (Appendix Table A1). Additionally, we used the variable of the social number calculated based on the multiple-choice answers of those participating in social activities. A sense of loneliness is part of the self-evaluation psychological health scale used in the CHARLS questionnaire, which has high scientific validity and accuracy.

#### 2.2.3. Other Control Variables

To better examine the significance level of the impact of explanatory variables on the dependent variables, this study controlled the impact of individual characteristic variables, such as gender, age, education level and marital status. In this study, although individuals aged 45 or above and 65 or below were strictly selected, taking into consideration the increase in the average life expectancy in rural areas and the phenomenon of incomplete retirement among rural residents, the age ceiling for migrant workers was extended to appropriately 70 years old. In the CHARLS questionnaire, education level was limited to the junior college level and below. It specifically included eight standards of educational attainment: illiterate, incomplete primary school, sishu/home school, elementary school, middle school, high school, vocational school, and two/three-year-college or associate degree. Marital status encompassed six specific categories: married with spouse present; married, but not living with spouse temporarily for reasons such as work; separated, divorced; widowed; and never married.

### 2.3. Descriptive Statistics and Nonparametric Test

All statistical analysis was performed using StataMP.17 software and descriptive statistics for the variables are presented in Table 2 and Table 3. The dependent variable, smoking, was assigned a value of 1 for smokers and 0 for non-smokers. The descriptive statistics showed that the proportion of migrant workers who smoked was 38.63% in 2013, 64.89% in 2015, and 62.57% in 2018, with smokers consuming an average of 20 cigarettes per day.

When the issue was analyzed via the work dimension, the income per month of migrant workers has exhibited a rising trend (Table 3), in line with the objective law of economic development. The average working time (including overtime) per day was around 8.5 h, which was slightly higher than the standard eight-hour working day. When the issue was considered from a social dimension (Table 2), participation in social activities was assigned a value of 1, and non-participation was assigned a value of 0. The statistics showed that nearly 60% of migrant workers participated in social activity. Moreover, the number of times that migrant workers participate in social activities per month was assigned a value, with participation in one activity assigned a value of 1, two activities assigned a value of 2, and three or more activities assigned a value of 3, with no value assigned otherwise. Overall, the number of migrant workers participating in social activities in the sample observations mostly ranged from 0 to 1 from 2013 to 2018, accounting for more than 40% and 30% of workers, respectively. It meant that the social activity level of migrant workers was low. When the issue was analyzed the life dimension, life satisfaction was assigned a value: not very satisfied was assigned a value of 1, quite satisfied was assigned a value of 2, very satisfied and above were assigned a value of 3, with no value assigned otherwise. The survey found that the life satisfaction of migrant workers ranged between quite satisfied and very satisfied, accounting for more than 90% of the cohort in 2013, 2015 and 2018. This meant that most migrant workers were satisfied with their existing lives. Meanwhile, the sense of loneliness that individuals felt was also assigned a value: rarely or none of the time was assigned a value of 0, some or a little was assigned a value of 1, occasional or moderate was assigned a value of 2, and most or all time was assigned a value of 3. Overall, more than 80% of migrant workers rarely felt lonely, and the proportion of those lonely most or all time was below 5% from 2013 to 2018.

With regard to control variables, more than 77% of migrant workers in the sample observation were male (Table 2), indicating a clear gender preference. Marital status was assigned a value ranging from 0–3 for six situations of married with spouse present, married but not living with spouse temporarily, separated, divorced, widowed and never married. The proportion of marriage (value = 2 and 3) was above 93% from 2013 to 2018. Education level referred to the highest level of education currently obtained and was assigned a value ranging from 0–4 for eight different levels including no formal education, did not finish primary school, sishu/home school, elementary school, middle school, high school, vocational school, and two/three-year-college or associate degree. More than 80% were educated in primary and middle school, which indicated that the education level of most migrant workers was not high.

After conducting data preprocessing, this study performed a chi-square independence test to examine the relationship between social activity, smoking behavior and gender. The left part of Table 4 indicated a significant correlation between social number and gender, with male migrant workers showing greater interest in and participating in more social activities per month than their female counterparts. The right part of Table 4 indicated a significant difference in smoking behavior between males and females. The current survey found that 76.28% of male and 6.27% of female migrant workers were smokers. Specifically, male migrant workers accounted for 97.11% of the smoking sample, while female migrant workers only represented 2.89%. These findings supported the expected difference in smoking rate between males and females.

### 2.4. Model Selection

Smoking behavior involved two indicators in this study, which were whether to smoke and smoking frequency. Smoking status was a binary dependent variable, indicating whether someone smoked or not, while smoking frequency referred to the consumption number of cigarettes. Obviously, non-smokers did not have a consumption number of cigarettes. Smoking behavior is clearly a two-stage decision-making process. In the first stage, it was necessary to identify whether the migrant worker had a smoking behavior, i.e., screening smoking samples. In the second stage, the smoking frequency of migrant workers who smoked was analyzed by examining the daily consumption number of cigarettes (how many cigarettes smoked per day). Therefore, this study chose to use the Heckman two-stage model to explore the influencing factors of smoking behavior among migrant workers.

The Heckman two-steps method was proposed by Heckman (1979) [32]. One of its strengths is that it can effectively address endogeneity problems caused by sample selection bias in empirical research, especially the problem of self-selection bias in sampling, which helps to make the research sample more representative. The model is divided into two steps. In the first step, a probit selection model is established for whether migrant workers smoke. The probability of migrant workers choosing to smoke is estimated using all the observation values in the total sample, and the inverse Mills ratio (IMR) is calculated for each observation value.
(1)$$\mathrm{Prob}(\gamma =1)=\phi (\beta_{0}+\sum_{i=1}^{n}\beta_{i}x_{i})$$
where Prob(⋅) represents the probability of an event occurring and refers to the probability of smoking behavior among migrant workers in the entire sample in this study. γ=1 represents the presence of smoking behavior, while γ=0 represents the absence of smoking behavior. ϕ(⋅) is the cumulative distribution function of the standard normal distribution, $\beta_{0}$ is a constant term, and $x_{i}$ represents the influence factors on smoking behavior among migrant workers $\beta_{i}$ is the corresponding estimated parameter, which reflects the influence degree that explanatory variables have on the dependent variable of smoking status. The corresponding inverse Mills ratio $\lambda_{i}$ is calculated using the parameter values obtained from Equation (1).
(2)$$\lambda_{i}=\varphi (\beta_{0}+\sum_{i=1}^{n}\beta_{i}x_{i})/\phi (\beta_{0}+\sum_{i=1}^{n}\beta_{i}x_{i})$$
where φ(⋅) refers to the probability density function of the standard normal distribution. If the value of $\lambda_{i}$ is non-zero and statistically significant, it indicates that there is sample selection bias. Because the inverse Mills ratio $\lambda_{i}$ is linearly related to the sample error and has a mean of 0, using the Heckman two-stage model can effectively correct for sample self-selection bias and help researchers to obtain unbiased estimates.

In the second step, the inverse Mills ratio (used to estimate the likelihood of selection bias variables) is included as a control variable in an ordinary least-squares regression model. It allows researchers to further analyze the effective variables that influence the smoking frequency of smokers using the sample of migrant workers who smoke.
(3)$$y=\alpha_{0}+\sum_{i=1}^{n}\alpha_{i}x_{i}+\omega_{i}\lambda_{i}+\varepsilon_{i}$$
where y represents smoking frequency, which refers to the number of cigarettes consumed per day. $x_{i}$ represents the influence factors on smoking frequency among migrant workers, while $\lambda_{i}$ represents the inverse Mills ratio calculated by Equation (2). $\alpha_{0}$ is the constant term, $\varepsilon_{i}$ is the random error term, and $\alpha_{i}$ and $\omega_{i}$ are the corresponding regression coefficients that reflect the degree to which the explanatory variables affect the smoking frequency.

## 3. Results and Analysis

### 3.1. Analysis of Preliminary Results

This study conducted the Heckman two-stage model to analyze its results using StataMP.17 software, and the results were presented in Table 5. There were four sets of model estimation results, in which models (1) and (2) were baseline regression results that covered the main independent variables, with the difference being whether or not control variables were added. On this basis, models (3) and (4) introduced the independent variable of work stress and distinguished between whether or not control variables were added for main effect testing. Furthermore, the inverse Mills ratios in models (1) and (3) were significantly different from 0 at the 10% level, indicating that sample selection bias impacted the model estimation in the first stage. With the addition of control variables, the inverse Mills ratios in models (2) and (4) were no longer significant, indicating that there was no selection bias problem in the sample. This further confirmed the suitability and rigor of using the Heckman two-stage model.

According to the estimation results (Table 5), the first stage of model (1) showed that income, social activity and life satisfaction had a significant positive impact on whether migrant workers smoke at levels of 1% and 10%, respectively, without considering control variables. This also indicated that, under certain conditions, a higher work income, higher life satisfaction, and participation in social activities could significantly increase the likelihood of smoking behavior among migrant workers. To some extent, smoking might be better attributed to the special social function it serves in Chinese culture: the rapid establishment of interpersonal relationships [10,33]. In the second stage of model (1), the independent variables of work income and life satisfaction had a significant negative impact on the smoking frequency of migrant workers at the 10% and 5% levels, respectively. This suggested that, as the living standards of migrant workers improved, their smoking frequency tended to decrease instead due to the increase in personal health awareness and more attention being paid to the negative effects of smoking and secondhand smoke on themselves and others. Under the premise of adding control variables, model (2) indicated that during the first stage, social activity, and sense of loneliness, as well as control variables of gender and age, all had positive impacts on smoking behavior among migrant workers, while life satisfaction was a key factor that negatively impacted smoking frequency in the second stage. This result not only confirmed the gender difference in smoking behavior among migrant workers [31], but also validated the existing conclusion that high loneliness and low life satisfaction can induce smoking behavior in the same context [34].

After introducing work stress as an independent variable, the regression results of model (3) compared to model (1) showed that in the first stage, the positive impact of work income, social activity, and life satisfaction on whether migrant workers smoke significantly increased, without considering control variables. In the second stage, the independent variables of income, social activity, and life satisfaction had significant negative impacts on smoking frequency among migrant workers at 10%, 5%, and 1% levels, respectively. With the control variables present, the regression results of model (4) showed that, in the first stage, the positive impact of social activity, sense of loneliness, and control variables of gender and age on whether migrant workers smoke were consistent with the model (2). During the second stage, the independent variables of life satisfaction and work stress, as well as the control variable of all levels of education, had significant negative impacts on smoking frequency. Specifically, the control variable education level had a negative impact at the 10% level, while the impact of the independent variable work stress was significantly less than 0 at the 1% level. This suggested that, as work stress increased, migrant workers faced more physical work and longer working times. These conditions could lead to social time tension and energy consumption, thus reduced the contact between migrant workers and smokers.

In summary, the independent variables of social activity and sense of loneliness had notable positive impacts on the first stage of the Heckman two-stage model, regardless of whether control variables were implemented. During the second stage, the independent variables of life satisfaction and work stress demonstrated a significant negative impact on smoking frequency among migrant workers. However, after accounting for other control variables, gender and age had noticeable positive impacts on whether migrant workers smoked, while the independent variable of income no longer had a significant impact on the two-stage decision-making process of smoking behavior. As work stress continued to be introduced as an independent variable, the control variable of education level had a significant negative impact on the smoking frequency of migrant workers. Additionally, it also strengthened the negative impact of the independent variable of life satisfaction on their smoking frequency.

### 3.2. Heterogeneity Analysis

In general, there were significant differences in economic development, social culture and customs among different regions, all of which could potentially have diverse effects on the two-stage process of smoking behavior among migrant workers. Considering the migration flow directions of migrant workers in China, this section examined the eastern region and central-western region as sub-samples on the basis of the models (2) and (4). The results are shown in Table 5 as controls to further break down the regional heterogeneity of smoking behavior. The regression results were shown in Table 6.

The sub-sample regression results from model (2) were inconsistent with the impacts revealed in the entire sample analysis (Table 6). Notably, in the first stage, the impacts of social activity and life satisfaction were considerably enhanced in the eastern region, whereas income and sense of loneliness had significant impacts in the central-western region. Furthermore, the control variables of gender and education level had significant effects in the central-western region, while the effects of the control variables in the eastern region were consistent with the effects of the entire sample. In the second stage, neither the independent variable of life satisfaction nor social activity exhibited significant effects in the eastern and central-western regions. However, the independent variable of social activity and the control variable of marital status demonstrated significant positive impacts in the central-western region. These findings implied that the two-stage process of smoking behavior of migrant workers in the eastern region was susceptible to significant impacts from social activity and life satisfaction, while the process in the central-western region was vulnerable to significant impacts from income.

Similarly, in the first stage, the sub-sample regression results from model (4), involving the independent variable of work stress, showed that, compared to the entire sample, the impacts of the independent variable of social activity and the control variables of gender and age were considerably enhanced in the eastern region, while the independent variable of life satisfaction showed a notable negative impact. In the central-western region, the impacts of the independent variables of income and sense of loneliness were significant, while the impact of social activity was not significant, and the impact effect of the control variable of education level was noteworthy. During the second stage, the independent variable of work stress had a significant negative impact on the smoking behavior of migrant workers in both the eastern and central-western regions, while life satisfaction did not show significant effects. However, the central-western region was still affected by social activity. In addition, the influence of control variables in the eastern region was consistent with the effects seen across the entire sample, but the central-western region was only significantly affected by marital status. Overall, the results were largely consistent with the sub-sample regression results of model (3), albeit without the inclusion of the independent variable of work stress (Table 6). The influencing factors in the eastern region were more related to mental aspects, such as social activity and interpersonal communication, while the factors in the central-western region were mainly affected by material conditions like income and consumption expenditures. Above all, these not only confirmed the regional heterogeneity of influencing factors in the two-stage process of smoking behavior among migrant workers in the eastern region and central-western region of China, but also reaffirmed the impacts of the key independent variables of social activity and work stress.

### 3.3. Robustness Test

Given that this study used the Heckman two-stage model to perform empirical analysis, the endogeneity issue with regards to sample selection had been eliminated. To further validate the robustness of the regression results mentioned above, this study also used Winsorization, core variable replacement and sub-sample regression methods. Among these methods, the sub-sample regression results are presented earlier in the heterogeneity analysis (Table 6) and are not repeated here.

Winsorization. This study used Winsorization to perform truncation correction on continuous variables by removing extreme values within the upper and lower 1% and 5%, respectively, in order to eliminate the biased influence of extreme values on the model’s regression results. The specific robust regression results could be found in model (5) and (6) in Table 7. Compared to model (2) in Table 5, the regression results remained consistent, regardless of whether 1% or 5% truncation correction was applied. Moreover, after eliminating the impact of extreme values, the regression coefficients of the main independent variables were obviously increased, and their influence effects were further strengthened. This indicated that the empirical analysis results of the Heckman two-stage model used in this study were robust enough.Core variable replacement. On the one hand, work stress and life stress were used as replacement variables for the variable of income in this paper. We chose to do this since life stress was a relative value of income and living expenses, while work stress involved working time and indirectly affected income. Model (7) in Table 7 presents the corresponding regression results, contrasting with model (4) in Table 5. It was found that the regression results obtained after rerunning the model had not changed significantly compared to before. On the other hand, social number was used as a replacement variable for the variable of social activity in order to re-measure the key independent variable of social interaction, as presented in model (8). This part aimed to examine whether the number of social activity attendance, another measure of social interaction, had an impact on the two-stage process of smoking behavior among migrant workers. It indicated that the results after re-regression were basically consistent with the previous ones. These findings supported the conclusion that social interaction and life satisfaction affected the two-stage process of smoking behavior among migrant workers, and also verified the robustness of the empirical analysis results of the Heckman two-stage model in this paper.

## 4. Discussion

### 4.1. Further Analysis of the Results

The empirical results of this study showed that social activity and sense of loneliness had significant positive impacts on whether Chinese migrant workers smoke, while work stress and life satisfaction had significant negative impacts on their smoking frequency. These results were consistent with previous studies [25,26,35,36]. However, some interesting findings have also emerged. For instance, under certain conditions, smoking behavior increased with more participation in social activities, whereas higher work stress resulted in lower smoking frequency. This association might be related to the socio-cultural characteristics of China [37].The Chinese live in a web of social relations determining the attainment and allocation of various resources. As such, it is a custom to gift and exchange cigarettes to express respect and hospitality [10,33]. Cigarettes may act as a social tie to rapidly build interpersonal relationships through a number of daily interactions and social occasions [38], thereby increasing the likelihood of contact with cigarettes. When work stress increased, it meant that migrant workers were faced with heavier physical labor and longer working time, which led to social time tension and energy consumption, thus reducing the contact between migrant workers and smokers. Strict no-smoking policies and smoking restrictions in the workplace were carried out. All these reduced the possibility of smoking, especially in social situations. Meanwhile, our analysis results confirmed the results of previous studies, i.e., Chinese men were more likely to smoke than women [34,39]. The gender difference was especially noticeable among migrant workers, which appeared to be closely related to national culture [10,40]. Furthermore, our empirical analysis also provided strong evidence in support of the view that smoking behavior is related to age and education level. Under certain circumstances, male migrant workers were more likely to smoke, and smoking rate increased with age. After reaching a certain stage, migrant workers with higher education levels were more likely to keep their smoking frequency to a lower level to reduce the health risks caused by smoking.

Inconsistent with previous findings in the literature [22,41], we found that work income can have different effects on the two-stage process of smoking behavior among migrant workers in China. Without adding the control variables, income significantly affected the two-stage process of smoking behavior. Once control variables were added, income no longer had a significant impact on smoking behavior. In fact, cigarette consumption has a threshold effect. Most migrant workers in China belong to the low-income group. As such, budget considerations may force individuals to reduce cigarette consumption or quit smoking altogether. However, after controlling for individual characteristics, the independent effect of work income was less evident or non-existent, suggesting that individuals might cope with work-related stress by smoking [42]. In addition, since smoking is an addictive behavior, income might not play a significant role in the consumption of addictive products due to dependence effects [11].

Furthermore, the results of our analysis in this study supplemented and broadened existing research. Regarding regional heterogeneity analysis, the primary factors influencing smoking behavior among migrant workers varied across different regions. Specifically, smoking behavior in the eastern region was mainly influenced by social activity and life satisfaction, while smoking was influenced by income in the central-western region. Overall, this highlighted the impact of psychological well-being in the former, and material conditions in the latter. These findings were consistent with current conditions in which the eastern region of China leads in economic development, social openness and spiritual civilization compared to the central-western region [16]. At the same time, our regression results in the central-western region indicated that marital status had a significant impact on smoking frequency among migrant workers, which provided clues for further research.

### 4.2. Limitations and Future Perspectives

Our interpretation of the regression results is constrained by certain limitations in this study. Firstly, the measurement of stress, including both work and life aspects, mainly relies on relative indicators and lacks the self-evaluation of work and life stress from migrant workers. Meanwhile, the CHARLS database is a typical longitudinal data source, and its variables are usually measured by a single item, meaning that the reliability and validity of the scale cannot be validated. Secondly, regional heterogeneity analysis in our study needs to be improved. Although we divided the entire sample into two groups, the eastern region and central-western region, respectively, the sample size between these two groups differed significantly, with more samples taken from the eastern region. This may impact the analysis results presented in this study. Thirdly, our original empirical logic aims to test the impact of different stress indicators on smoking behavior (whether to smoke and smoking frequency) by sequentially adding the variables of work stress and life stress as determined based on the benchmark model. However, because the measurement of life stress in this paper is based on the relative value of income and expenses, there is collinearity between income and life stress. As such, taking the life stress variable as a part of robustness test of the model also better satisfies our initial research hypothesis. Additionally, smoking history (the number of years of smoking) may be a potential factor affecting the smoking frequency of migrant workers.

Population aging currently poses a great challenge to the family and to the social public service system. The Chinese migrant worker population is also experiencing aging, and there is a phenomenon of intergenerational shift occurring within the group whereby the new generation of migrant workers, especially those born in the 1990s and 2000s, are gradually becoming the main force in urban development. Due to the different influences of growth environment, educational and cultural background, the new generation of migrant workers differs significantly from their older counterparts in terms of value pursuit, behavior, consumption habits and so on. Furthermore, the topic of retirement and re-employment for migrant workers who have exceeded the typical retirement age has also been widely discussed by society. These are essential aspects of social life that urgently require more attention in future research.

## 5. Conclusions

As the mortality attributable to smoking continued to increase, tobacco smoking will remain a major public health problem in most Asian countries in the coming decades [43]. Migrant workers in China, as a marginalized and vulnerable group, are a unique product of the dual urban–rural system structure. Given that population aging continues to spread, it is of great practical significance to pay more attention to the public health problems that affect migrant workers, including smoking behavior. In this study, based on panel data from the CHARLS database from 2013 to 2018, the Heckman two-stage model was used to empirically analyze the impact of social interaction, work stress and life stress on smoking behavior (whether to smoke and smoking frequency) among Chinese migrant workers.

Firstly, the empirical analysis in this study indicated that social interaction and a sense of loneliness were found to have a significant positive impact on whether Chinese migrant workers smoke, while work stress and life satisfaction had a significant negative impact on the smoking frequency of those. Therefore, in the context of urban–rural integration and development, it is vital to strengthen social integration in order to improve life satisfaction and reduce sense of loneliness. Such an approach is crucial for controlling smoking behavior among migrant workers and, if implemented, would have a positive impact.

Secondly, smoking behavior among migrant workers was also influenced by factors such as gender, age, and education level. The gender difference in smoking behavior among Chinese migrant workers was particularly notable. We also found that the smoking rate among migrant workers increased with age, but that those with a higher level of education were more likely to keep their smoking frequency to a lower level. Therefore, it is essential to strengthen anti-smoking propaganda and promote the development of vocational education and training in rural areas. Such initiatives would raise awareness about the harmful effects of smoking among rural minors and enhance the overall education level and quality of migrant workers.

Finally, this study also examined the regional heterogeneity of smoking behavior among Chinese migrant workers. The results showed that in the eastern region of China, smoking behavior was primarily influenced by psychological well-being factors such as social activity and life satisfaction. Conversely, smoking behavior was more significantly influenced by material conditions such as income in the central-western region. Of course, more scientific research is necessary to further validate these findings.

## Figures and Tables

**Figure 1 behavsci-13-00680-f001:**
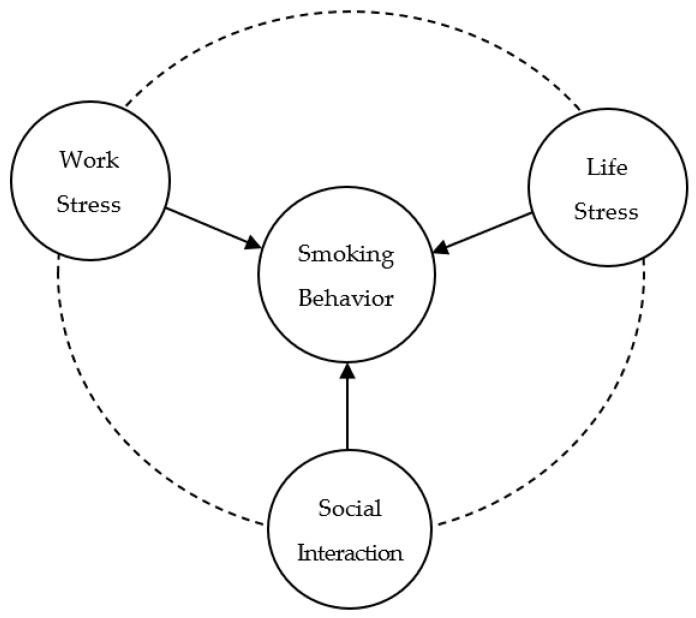
Main concepts and framework structure.

**Table 1 behavsci-13-00680-t001:** Variables from China Health and Retirement Longitudinal Study (CHARLS) Data.

Variable	Definition	CHARLS Code
Dependent variable:		
Smoking	Smoking behavior	da059
Smoking frequency	How many cigarettes per day	da063
Independent variable:		
Work dimension		
Income	Income per month from employment (yuan)	ff002~ff009
Income per hour	Converted into income per hour (yuan) ^①^	——
Working time	Working time including overtime per day	fe003
Work stress	Relative index of work intensity per day	——
Life dimension		
Life satisfaction	How satisfied you are with your live	dc028
Life stress	Relative index of income and basic living expenses ^②^	——
Social interaction		
Social activity	Take part in social activities	da056
Social number	How many social activities per week	——
Sense of loneliness	Degree to which you feel lonely	dc017
Control variable:		
Gender	Male or female	ba000_w2_3
Age	Years	zba002_1
Education	The highest education received	zbd001
Marriage	Marital status	be001

Notes: ① Unified caliber, calculated workdays of 26 per month according to the common practices and work reality of migrant workers. ② Refer to global poverty line, the latest standards updated is USD 2.15 issued by World Bank in 2022, it is converted into CNY 15.447 [27].

**Table 2 behavsci-13-00680-t002:** Descriptive statistics of categorical variables in 2013, 2015 and 2018.

Categorical Variable	Value Description		Year	Freq. (n)	Pct. (%)	Year	Freq. (n)	Pct. (%)	Year	Freq. (n)	Pct. (%)
Smoking	1 = smoking, 0 = no smoking	0	2013	143	61.37	2015	132	35.11	2018	131	37.43
		1	2013	90	38.63	2015	244	64.89	2018	219	62.57
Social activity	1 = yes, 0 = no participating	0	2013	152	43.68	2015	153	41.46	2018	172	44.33
		1	2013	196	56.32	2015	216	58.54	2018	216	55.67
Social number	0 = none, 1 = one term, 2 = two terms, 3 = three or more items	0	2013	152	43.68	2015	153	41.46	2018	172	44.33
		1	2013	105	30.17	2015	130	35.23	2018	120	30.93
		2	2013	54	15.52	2015	54	14.63	2018	55	14.18
		3	2013	37	10.63	2015	32	8.67	2018	41	10.57
Life satisfaction	0 = not at all satisfied, 1 = not very satisfied, 2 = somewhat satisfied, 3 = very and completely satisfied	0	2013	4	1.16	2015	1	0.27	2018	3	0.81
		1	2013	31	9.01	2015	13	3.54	2018	22	5.95
		2	2013	240	69.77	2015	189	51.50	2018	211	57.03
		3	2013	69	20.96	2015	164	44.69	2018	134	36.22
Sense of loneliness	0 = rarely or none of the time, 1 = some or a little, 2 = occasional or moderate, 3 = most or all time	0	2013	290	84.80	2015	310	84.24	2018	301	81.35
		1	2013	30	8.77	2015	28	7.61	2018	35	9.46
		2	2013	13	3.80	2015	23	6.25	2018	16	4.32
		3	2013	9	2.63	2015	7	1.90	2018	18	4.86
Gender	1 = male, 0 = female	0	2013	84	21.65	2015	87	22.42	2018	86	22.16
		1	2013	304	78.35	2015	301	77.58	2018	302	77.84
Education	0 = no formal education, 1 = did not finish primary school, home school, elementary school, 2 = middle school, 3 = high school, vocational school, 4= two/three-year-college or associate degree	0	2013	25	6.44	2015	25	6.44	2018	25	6.44
		1	2013	147	37.89	2015	147	37.89	2018	147	37.89
		2	2013	148	38.14	2015	148	38.14	2018	148	38.14
		3	2013	64	16.49	2015	64	16.49	2018	64	16.49
		4	2013	4	1.03	2015	4	1.03	2018	4	1.03
Marriage	0 = never married, 1 = separated, divorced, widowed, 2 = married but not living with spouse temporarily, 3 = married with spouse present	0	2013	2	0.52	2015	2	0.52	2018	2	0.52
		1	2013	15	3.87	2015	20	5.15	2018	23	5.93
		2	2013	41	10.57	2015	43	11.08	2018	43	11.08
		3	2013	330	85.05	2015	323	83.25	2018	320	82.47
Observations	Survey sample of migrant workers from three follow-up waves	388									

Notes: Freq. represents the corresponding frequency; Pct. represents the percentage.

**Table 3 behavsci-13-00680-t003:** Descriptive statistics of continuous variables in 2013, 2015 and 2018.

Continuous Variable	Value Description	Year	Mean	Std. DEV.	Min	Max
Smoking frequency	Number of cigarettes smoked per day (cigarettes)	2013	19.311	13.312	1	60
		2015	20.033	12.828	1	70
		2018	20.516	13.443	1	80
Income	Income per month from employment (yuan)	2013	2563.339	2106.732	400	23,000
		2015	2669.041	1607.887	460	12,000
		2018	3044.851	1943.985	400	15,000
Working time	Working time including overtime per day (hour)	2013	8.723	1.845	2	12
		2015	8.482	1.942	1	12
		2018	8.491	1.963	1	12
Work stress	Work intensity per day: (working hours)/(standard hours)	2013	1.090	0.231	0.250	1.5
		2015	1.060	0.243	0.125	1.5
		2018	1.061	0.245	0.125	1.5
Income per hour	Income per hour: income per month/26/working time per day	2013	11.694	8.601	2.25	76.67
		2015	13.299	8.396	1.67	60.00
		2018	15.540	12.295	1.67	120.00
Life stress	Life stress per day: (income per day)/(the latest global poverty standard by Word Bank in 2022)	2013	6.638	5.455	1.036	59.558
		2015	6.911	4.164	1.191	31.074
		2018	7.885	5.034	1.036	38.842
Age	Years	2013	53.227	5.045	45	65
		2015	55.227	5.045	47	67
		2018	58.227	5.045	50	70
Observations	Survey sample of migrant workers from three follow-up waves	388				

**Table 4 behavsci-13-00680-t004:** Chi-square test of social interaction, smoking behavior and gender.

Gender	Social Number					Smoking Behavior		
	0	1	2	3	Total	0	1	Total
0 = female	114	90	29	13	246	239	16	255
1 = male	363	265	134	97	859	167	537	704
Total	477	355	163	110	1105	406	553	959
Test parameter	Pearson $\chi^{2}$ = 11.51, Prob = 0.009					Pearson $\chi^{2}$ = 375.77, Prob < 0.001		

**Table 5 behavsci-13-00680-t005:** Regression results of Heckman two-stage model.

Variables	(1)		(2)		(3)		(4)	
	First	Second	First	Second	First	Second	First	Second
Income	0.000 (>0) *** (4.040)	−0.002 ^†^ (−1.841)	0.000 (>0) (0.406)	−0.000 (>0) (−0.573)	0.000 (>0) *** (3.948)	−0.002 ^†^ (−1.741)	0.000 (>0) (0.437)	−0.000 (>0) (−0.166)
Social activity	0.185 ^†^ (1.841)	−0.189 (−0.069)	0.252 * (1.968)	6.482 (1.408)	0.192 ^†^ (1.886)	−0.799 (−0.293)	0.253 ^†^ (1.955)	5.007 (1.045)
Life satisfaction	0.141 ^†^ (1.653)	−4.887 * (−2.215)	−0.043 (−0.399)	−2.648 ^†^ (−1.892)	0.145 ^†^ (1.697)	−5.140 * (−2.359)	−0.044 (−0.408)	−2.762 ^†^ (−1.960)
Sense of loneliness	0.075 (0.965)	−1.981 (−1.372)	0.201 ^†^ (1.820)	1.426 (0.472)	0.067 (0.857)	−1.444 (−1.062)	0.200 ^†^ (1.801)	1.191 (0.383)
Work stress	——	——	——	——	0.146 (0.688)	−12.180 *** (−3.530)	−0.077 (−0.299)	−11.185 *** (−3.597)
Gender	——	——	2.692 *** (12.376)	56.691 (0.746)	——	——	2.698 *** (12.330)	40.595 (0.516)
Age	——	——	0.023 ^†^ (1.955)	0.066 (0.157)	——	——	0.023 ^†^ (1.885)	−0.068 (−0.161)
Education	——	——	0.073 (0.928)	−1.679 (−1.154)	——	——	0.064 (0.815)	−2.488 ^†^ (−1.806)
Marriage	——	——	−0.144 (−1.097)	−0.278 (−0.113)	——	——	−0.137 (−1.046)	−0.068 (−0.028)
IMR	——	−37.983 * (−1.706)	——	21.245 (0.536)	——	−37.759 * (−1.770)	——	12.014 (0.292)
_cons	−0.527 ** (−2.394)	64.089 *** (2.678)	−3.052 *** (−3.546)	−40.385 (−0.361)	−0.695 ** (−2.212)	77.067 *** (3.053)	−2.939 *** (−3.216)	0.118 (0.001)
N	658	392	658	392	653	388	653	388

Notes: First and Second are the estimation results of the first and second steps of the model respectively; 0.000 (>0) represents approximate values for non-zero coefficients; t statistics in parentheses; † *p* < 0.1, * *p* < 0.05, ** *p* < 0.01, *** *p* < 0.001; IMR represents the inverse Mills ratio used to estimate the likelihood of selection bias variables; _cons represents the constant term; N represents the number of samples.

**Table 6 behavsci-13-00680-t006:** Regression results of Heckman two-stage model by region for Model (2) and (4).

Variables	(2)- Eastern Region		(2)- Central-Western		(4)- Eastern Region		(4)- Central-Western	
	First	Second	First	Second	First	Second	First	Second
Income	−0.000 (>0) (−0.727)	−0.000 (>0) (−0.680)	0.000 (>0) ** (2.611)	0.001 (0.531)	−0.000 (>0) (−0.472)	−0.000 (>0) (−0.824)	0.000 (>0) * (2.458)	0.001 (0.661)
Social activity	0.486 ** (2.899)	1.039 (0.128)	−0.053 (−0.229)	4.907 ^†^ (1.907)	0.459 ** (2.694)	6.776 (0.887)	−0.022 (−0.095)	4.539 ^†^ (1.806)
Life satisfaction	−0.276 ^†^ (−1.950)	−0.186 (−0.041)	0.289 (1.560	−0.861 (−0.258)	−0.270 ^†^ (−1.902)	−3.714 (−0.833)	0.290 (1.541)	−1.274 (−0.402)
Sense of loneliness	0.132 (0.901)	−1.171 (−0.501)	0.365 ^†^ (1.955)	2.047 (0.624)	0.154 (1.037)	0.626 (0.242)	0.342 ^†^ (1.805)	2.785 (0.927)
Work stress	——	——	——	——	−0.351 (−1.022)	−11.284 ^†^ (−1.717)	0.410 (0.956)	−9.910 ^†^ (−1.683)
Gender	2.864 *** (9.732)	−17.379 (−0.237)	2.534 *** (6.819)	42.734 (0.984)	2.871 *** (9.683)	41.977 (0.577)	2.494 *** (6.700)	50.364 (1.193)
Age	0.041 * (2.517)	−0.448 (−0.667)	0.010 (0.520)	−0.050 (−0.211)	0.041 * (2.486	0.079 (0.118)	0.010 (0.494)	−0.158 (−0.668)
Education	−0.034 (−0.334)	−1.280 (−1.107)	0.253 ^†^ (1.831)	−2.241 (−0.853)	−0.058 (−0.551)	−2.356 ^†^ (−1.679)	0.256 ^†^ (1.842)	−2.421 (−0.959)
Marriage	−0.255 (−1.530)	0.472 (0.127)	0.140 (0.560)	6.446 * (2.200)	−0.244 (−1.466)	−2.486 (−0.690)	0.170 (0.679)	6.071 * (2.018)
IMR	——	−15.642 (−0.421)	——	13.692 (0.572)	——	14.196 (0.383)	——	15.918 (0.687)
_cons	−3.245 ** (−2.841)	72.756 (0.670)	−4.224 ** (−2.919)	−40.856 (−0.549)	−2.887 * (−2.366)	−1.834 (−0.018)	−4.657 ** (−2.964)	−30.312 (−0.399)
N	416	226	242	166	413	223	240	165

Notes: First and Second are the estimation results of the first and second steps of the model respectively; 0.00(>0) represents approximate values for non-zero coefficients; t statistics in parentheses; † *p* < 0.1, * *p* < 0.05, ** *p* < 0.01, *** *p* < 0.001; IMR represents the inverse Mills ratio used to estimate the likelihood of selection bias variables; _cons represents the constant term; N represents the number of samples.

**Table 7 behavsci-13-00680-t007:** Regression results of model robustness test.

Variables	(5)		(6)		(7)		(8)	
	First	Second	First	Second	First	Second	First	Second
Income	0.000 (>0) (0.606)	−0.000 (>0) (−0.016)	0.000 (>0) (0.986)	0.000 (>0) (0.311)	——	——	0.000 (>0) (0.498)	−0.000 (>0) (−0.679)
Social activity	0.252 * (1.969)	6.170 (1.399)	0.255 * (1.991)	3.896 (1.111)	0.253 ^†^ (1.955)	5.007 (1.045)	——	——
Social number	——	——	——	——	——	——	0.115 ^†^ (1.788)	1.928 (0.884)
Life satisfaction	−0.046 (−0.418)	−2.664 ^†^ (−1.956)	−0.050 (−0.457)	−2.344 * (−2.089)	−0.044 (−0.408)	−2.762 ^†^ (−1.960)	−0.047 (−0.427)	−2.369 ^†^ (−1.632)
Sense of loneliness	0.201 ^†^ (1.818)	1.543 (0.535)	0.201 ^†^ (1.814)	1.155 (0.511)	0.200 ^†^ (1.801)	1.191 (0.383)	0.197 ^†^ (1.799)	0.269 (0.083)
Work stress	——	——	——	——	−0.077 (−0.299)	−11.185 *** (−3.597)	——	——
Life stress	——	——	——	——	0.006 (0.437)	−0.029 (−0.166)	——	——
Gender	2.685 *** (12.321)	56.859 (0.785)	2.672 *** (12.247)	37.765 (0.668)	2.698 *** (12.330)	40.595 (0.516)	2.690 *** (12.329)	23.577 (0.286)
Age	0.024 * (1.977)	0.083 (0.206)	0.024 * (2.009)	−0.026 (−0.082)	0.023 ^†^ (1.885)	−0.068 (−0.161)	0.022 ^†^ (1.847)	−0.129 (−0.305)
Education	0.073 (0.929)	−1.494 (−1.071)	0.072 (0.915)	−1.307 (−1.198)	0.064 (0.815)	−2.488 ^†^ (−1.806)	0.063 (0.789)	−2.390 (−1.688)
Marriage	−0.144 (−1.083)	−0.145 (−0.062)	−0.140 (−1.066)	0.225 (0.124)	−0.137 (−1.046)	−0.068 (−0.028)	−0.137 (−1.046)	0.745 (0.297)
IMR	——	21.496 (0.567)	——	12.452 (0.419)	——	12.014 (0.292)	——	3.832 (0.089)
_cons	−3.083 *** (−3.579)	−42.967 (−0.401)	−3.143 *** (−3.645)	−16.600 (−0.196)	−2.939 ** (−3.216)	0.118 (0.001)	−2.936 *** (−3.441)	10.295 (0.087)
N	658	392	658	392	653	388	658	392

Notes: First and Second are the estimation results of the first and second steps of the model respectively; 0.000 (>0) represents approximate values for non-zero coefficients; t statistics in parentheses; † *p* < 0.1, * *p* < 0.05, ** *p* < 0.01, *** *p* < 0.001; IMR represents the inverse Mills ratio used to estimate the likelihood of selection bias variables; _cons represents the constant term; N represents the number of samples.

## Data Availability

CHARLS database is open to public for research purposes. The data can be accessed and downloaded from the CHARLS homepage (URL: http://charls.pku.edu.cn/en, accessed on 10 October 2022).

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
