# Peer review of "Social Interaction, Survival Stress and Smoking Behavior of Migrant Workers in China—An Empirical Analysis Using CHARLS Data from 2013–2018"

_behavsci, 2023, doi:10.3390/bs13080680_

Round 1

Author Response

Thank you very much for making very kind and thoughtful comments. We have revised our manuscript based on the comments from the reviewers. We hope sincerely that our revision and responses are satisfactory.

Reviewer 2 Report

The paper is well written and I only have some minor revisions.

1. Page 1 of 18; Abstract; Line number : 14-16

" Results showed that social activity and sense of loneliness had a significant positive impact on whether one smoked, while work stress and life satisfaction had a significant negative impact on their smoking frequency."

Authors may want to rephrase and clarify this sentence because I am unsure whether the positive and negative impacts mentioned refer to an increase or decrease in smoking frequency.

2. Page 4 of 18; Line number: 150-151

"After removing unmatched data and missing samples"

Authors may want to explain or clarify what the term "unmatched data" refers to.

3. Page 6 of 18

Table 2. 

Smoking is a binary variable, so maybe add frequency (n) and percentage (%) instead of mean, std, min or max in the Table 2. 

English is acceptable. 

Author Response

Thank you very much for your comment and suggestion. The paper is well written and I only have some minor revisions.

  1. Page 1 of 18; Abstract; Line number: 14-16

" Results showed that social activity and sense of loneliness had a significant positive impact on whether one smoked, while work stress and life satisfaction had a significant negative impact on their smoking frequency."

Authors may want to rephrase and clarify this sentence because I am unsure whether the positive and negative impacts mentioned refer to an increase or decrease in smoking frequency.

Response: Thank you very much for your comment and suggestion. We restated this sentence: “Smoking rates of migrant workers were positively associated with participating in social activity and sense of loneliness, while smoking frequency were negatively associated with work stress and life satisfaction.” We hope that the restated content is satisfactory, and easier for readers to understand.

  1. Page 4 of 18; Line number: 150-151

"After removing unmatched data and missing samples"

Authors may want to explain or clarify what the term "unmatched data" refers to.

Response: Thank you very much for your comment. Our research sample is migrant workers group. Combining with the data provided by the CHARLS database, a large number of samples in the CHARLS were screened according to matching criteria such as education level, household registration (hukou) and job nature. Based on the sample of migrant workers in 2013, the data of 2015 and 2018 were matched, and finally the panel data set for three consecutive waves was obtained. Please see on the line 160-161 of Pag 4.

  1. Page 6 of 18 Table 2.

Smoking is a binary variable, so maybe add frequency (n) and percentage (%) instead of mean, std, min or max in the Table 2.

Response: Thank you very much for making very kind and thoughtful comments. We fully agree with the reviewer's opinion, combined with the main variables in this paper, we split the original Table 2 into Table 2 of categorical variable and Table 3 of continuous variable in our revised manuscript. Please see on the Table 2 and 3 of the revised manuscript.

Once again, special thanks to you for your good comments.Please see the attachment.

Reviewer 3 Report

1. Lines 87 -88: higher SES smoked more? higher work intensity was associated with lower smoking level? Please double check, as they do not sound correct.

2. The background did not discuss the gender difference, but the purpose of this study mentioned it. It is better to talk about it in background as well.

3. Lines 148 - 152: It is unclear why only older and lower education level sample were included. Please provide rationale for the inclusion criteria mentioned in lines 148-152.

4. It is unclear if the analyses being conducted have took into account the complex survey sampling methods used in CHARLS data.

5. Table 2. It will be unclear to the readers why each variable has 3 rows. Please label the years. Also, It is inappropriate to present Mean/SD for binary or categorical variable such as smoking status and education level.

6. Table 3. Please directly label gender 0 and 1 in Table 3. Male and female? Present %s in Table 3 as well.

7.  The flow of this study is unusual. The analyses and results were all presented Materials and Methods section. Occasionally, the authors even talked about previous studies (e.g., lines 250-253) which are usually presented in Discussion section.

8. Table 4. It is unclear what are "imr" and "_cons". Please justify why 0.1 was used as one of the significant levels.

9. The examination of region difference was not mentioned in background nor study purpose.

10.  The section of conclusions is long. The authors could consider shortening the conclusion section.

NA

Author Response

  1. Lines 87 -88: higher SES smoked more? higher work intensity was associated with lower smoking level? Please double check, as they do not sound correct.

Response: Thank you very much for your comment and suggestion. We re-examined the expression of this sentence according to the Reference [24]. At the same time, we also added the existing research literature on socioeconomic status (SES) widely regarded as being an important determinant of smoking behavior and adjusted the sentence semantics. I hope our revised manuscript is satisfactory.

  1. The background did not discuss the gender difference, but the purpose of this study mentioned it. It is better to talk about it in background as well.

Response: Thank you for your valuable suggestion. The gender difference content has been added to the background. Please see on line 71 and 91 of Page 2.

  1. Lines 148 - 152: It is unclear why only older and lower education level sample were included. Please provide rationale for the inclusion criteria mentioned in lines 148-152.

Response: Thank you very much for your comment. First of all, due to the age limit of the survey samples in the CHARLS database, as mentioned in the section of 2.1 Data sources, “The CHARLS is a set of high-quality micro-survey data comprised of Chinese adults aged 45 and over and their family members for scientific research”. At the same time, combined with the migration direction of migrant workers, this paper mainly focuses on rural to urban areas, and according to the requirements of bachelor degree or above in the talent introduction policy of local governments, this study finally determines the inclusion criteria of migrant workers. We hope our response is clear and satisfactory.

  1. It is unclear if the analyses being conducted have took into account the complex survey sampling methods used in CHARLS data.

Response: Thank you very much for your comment. As mentioned in the section of 2.1 Data sources on line 142-144 and 153-157 of Page 4. “The baseline national wave of CHARLS was conducted in 2011 by the Social Science Research Center of Peking University, and a stratified (by per capita GDP of urban districts and rural counties) multi-stage (county/district-village/community household) PPS random sampling strategy was adopted.” “One of the many attractive features of this dataset, which makes it particularly suitable for our empirical analysis, is that it provides not only rich sample information such as demographic background, employment status, income, health status and other contents but also provides specific variable information on closely related to smoking behavior and social interaction.”

  1. Table 2. It will be unclear to the readers why each variable has 3 rows. Please label the years. Also, It is inappropriate to present Mean/SD for binary or categorical variable such as smoking status and education level.

Response: Thank you very much for your comment and suggestion. We fully agree with the reviewer's opinion! Combined with other reviewers' suggestions, we added the year column in Table 2 and Table 3 to make it easier for readers to understand. According to the main variables in this paper, we split the original Table 2 into Table 2 of categorical variable and Table 3 of continuous variable in our revised manuscript. Please see on the Table 2 and 3 of the revised manuscript.

  1. Table 3. Please directly label gender 0 and 1 in Table 3. Male and female? Present %s in Table 3 as well.

Response: Thank you for pointing this out. We have modified the expressions of “1=Male” and “0=female”. Please see the Table 4 of Page 8.

  1. The flow of this study is unusual. The analyses and results were all presented Materials and Methods section. Occasionally, the authors even talked about previous studies (e.g., lines 250-253) which are usually presented in Discussion section.

Response: Thank you for your valuable suggestion. In the revised manuscript, we have removed the contents that is not relevant to the section of Result and Analysis and moved some content of the analysis to the Discussion section. Please see on line 268-269, 348-350, 422-423 and 486-487. We hope our revised manuscript is clear and satisfactory.

  1. Table 4. It is unclear what are "imr" and "_cons". Please justify why 0.1 was used as one of the significant levels.

Response: Thank you for your good comments and useful suggestions. Combining the Model Selection of 2.4 and Equation (2) in this paper, we uniformly adjust "imr" to "IMR" to facilitate readers' understanding. At the same time, "_cons" is marked to explain its contents. Please see on line 290, and Table 5-7 of the revised manuscript.

Because the CHARLS is a set of high-quality micro-survey data comprised of Chinese adults aged 45 and over and their family members for scientific research, this may lead to insufficient accuracy of the research data. In addition, referring to the research literature using the CHARLS database and the existing literature on the smoking behavior of migrant workers as followed, we found a high significant level (0.1) was usually used:

(1). Pan, Z. Socioeconomic Predictors of Smoking and Smoking Frequency in Urban China: Evidence of Smoking as a Social Function. Health Promot Int 2004, 19, 309–315, doi:10.1093/heapro/dah304.

(2). Liu, J.; Rozelle, S.; Xu, Q.; Yu, N.; Zhou, T. Social Engagement and Elderly Health in China: Evidence from the China Health and Retirement Longitudinal Survey (CHARLS). Int J Environ Res Public Health 2019, 16, 278, doi:10.3390/ijerph16020278.

(3). Zhang, C.; Lei, X.; Strauss, J.; Zhao, Y. Health Insurance and Health Care among the Mid-Aged and Older Chinese: Evidence from the National Baseline Survey of CHARLS. Health Econ 2017, 26, 431–449, doi:10.1002/hec.3322.

(4). Florkowski, W.J.; Liu, Z.; Chen, H. Social Isolation: A Key to Explain a Migrant Worker Cigarette Smoking. Journal of the Asia Pacific Economy 2022, 0, 1–13, doi:10.1080/13547860.2021.2024362.

(5). Hao, X.; Yang, Y.; Gao, X.; Dai, T. Evaluating the Effectiveness of the Health Management Program for the Elderly on Health-Related Quality of Life among Elderly People in China: Findings from the China Health and Retirement Longitudinal Study. Int J Environ Res Public Health 2019, 16, 113, doi:10.3390/ijerph16010113.

(6). Weyers, S.; Dragano, N.; Möbus, S.; Beck, E.-M.; Stang, A.; Möhlenkamp, S.; Jöckel, K.H.; Erbel, R.; Siegrist, J. Poor Social Relations and Adverse Health Behaviour: Stronger Associations in Low Socioeconomic Groups? Int J Public Health 2010, 55, 17–23, doi:10.1007/s00038-009-0070-6.

Please refer to the above literatures.

  1. The examination of region difference was not mentioned in background nor study purpose.

Response: Thank you for your careful comment and kind suggestion. We have supplemented the content of regional differences in the background and study purpose. Please see on line 91, 108 and 132 in the latest revised manuscript.

  1. The section of conclusions is long. The authors could consider shortening the conclusion section.

Response: Thank you very much for your suggestion. We have revised and adjusted the relevant contents. In fact, the conclusion section of this paper not only contains brief research conclusions, but also some suggestions on tobacco control for migrant workers.

Once again, special thanks to you for your good comments. We sincerely hope this revision is satisfactory.
